# Temporal trends of severity and outcomes of critically ill patients with COVID-19 after the emergence of variants of concern: A comparison of two waves

Daniela Helena Machado Freitas[1]*, Eduardo Leite Vieira Costa[1], Natalia Alcantara Zimmermann[1], Larissa Santos Oliveira Gois[1], Mirella Vittig Alves Anjos[1], Felipe Gallego Lima[2], Pâmela Santos Andrade[3], Daniel Joelsons[4], Yeh-Li Ho[4], Flávia Cristina Silva Sales[4,5], Ester Cerdeira Sabino[4,5], Carlos Roberto Ribeiro Carvalho[1], Juliana Carvalho Ferreira[1]

1 Divisao de Pneumologia, Faculdade de Medicina, Instituto do Coracao (InCor), Hospital das Clinicas HCFMUSP, Universidade de Sao Paulo, Sao Paulo, SP, Brazil, 2 Divisao de Cardiologia, Faculdade de Medicina, Instituto Do Coracao (InCor), Hospital das Clinicas HCFMUSP, Universidade de Sao Paulo, Sao Paulo, SP, Brazil, 3 Departamento de Epidemiologia, Faculdade de Saude Publica, Universidade de Sao Paulo, Sao Paulo, Brazil, 4 Divisao de Clinica de Molestias Infecciosas e Parasitarias, Departamento de Molestias Infecciosas e Parasitarias, Faculdade de Medicina, Hospital das Clinicas HCFMUSP, Universidade de Sao Paulo, Sao Paulo, Brazil, 5 Faculdade de Medicina, Instituto de Medicina Tropical, Universidade de Sao Paulo, Sao Paulo, Brazil

* daniela.f@hc.fm.usp.br

**Data Availability Statement:** All relevant data are within the paper and its Supporting Information files. Brazilian Data privacy regulations prohibit

## Abstract

### Background

The emergence of SARS-CoV-2 variants led to subsequent waves of COVID-19 worldwide. In many countries, the second wave of COVID-19 was marked by record deaths, raising the concern that variants associated with that wave might be more deadly. Our aim was to compare outcomes of critically-ill patients of the first two waves of COVID-19.

### Methods

This retrospective cohort included critically-ill patients admitted between March-June 2020 and April-July 2021 in the largest academic hospital in Brazil, which has free-access universal health care system. We compared admission characteristics and hospital outcomes. The main outcome was 60-day survival and we built multivariable Cox model based on a conceptual causal diagram in the format of directed acyclic graph (DAG).

### Results

We included 1583 patients (1315 in the first and 268 in the second wave). Patients in the second wave were younger, had lower severity scores, used prone and non-invasive ventilatory support more often, and fewer patients required mechanical ventilation (70% vs 80%, $p<0.001$), vasopressors (60 vs 74%, $p<0.001$), and dialysis (22% vs 37%, $p<0.001$). Survival was higher in the second wave (HR 0.61, 95%CI 0.50–0.76). In the multivariable

sharing of individual level data to the public and the ethical approval did not cover public sharing of data for unknown purposes. Anonymized data may be shared upon reasonable request upon contact with the corresponding author (daniela.f@hc.fm.usp.br) or the Research Ethics Committee (cappesq.adm@hc.fm.usp.br).

**Funding:** The author(s) received no specific funding for this work.

**Competing interests:** Dr. Ferreira reports personal fees from Medtronic, outside the submitted work; Dr. Costa reports personal fees from Timpel, personal fees from Magnamed, outside the submitted work; Dr. Ho reports personal fees from Pan-American Health Organization, outside the submitted work. This does not alter our adherence to PLOS ONE policies on sharing data and materials. The other authors have no conflict of interest to disclose.

**Abbreviations:** aHR, Adjusted hazard ratio; ARDS, Acute Respiratory Distress Syndrome; BMI, Body mass index; COVID-19, Coronavirus Disease 2019; DAG, Directed acyclic graph; FiO2, Fraction of inspired oxygen; HCFMUSP, Hospital das Clínicas da Faculdade de Medicina da Universidade de São Paulo; HFNC, High-flow nasal cannula; HR, Hazard ratio; IBW, Ideal body weight; ICU, Intensive care unit; IgG, Immunoglobulin G; IgM, Immunoglobulin M; IQR, Interquartile range; NIV, Noninvasive ventilation; PaO2/FiO2, Partial pressure of arterial oxygen and fraction of inspired oxygen ratio; PEEP, Positive end-expiratory pressure; REDCap, Research Electronic Data Capture; RRT, Renal replacement therapy; RT-PCR, Reverse Transcription Polymerase Chain Reaction; SAPS 3, Simplified Acute Physiology Score III; SARI, Severe acute respiratory infection; SARS-CoV-2, Severe Acute Respiratory Syndrome—Coronavirus 2; SD, standard deviation; SOFA, Sequential (Sepsis-related) Organ Failure Assessment; STROBE, Strengthening The Reporting of Observational Studies in Epidemiology; VOCs, Variants of concern.

model, admission during the second wave, adjusted for age, SAPS3 and vaccination, was not associated with survival (aHR 0.85, 95%CI 0.65–1.12).

## Conclusions

In this cohort study, patients with COVID-19 admitted to the ICU in the second wave were younger and had better prognostic scores. Adjusted survival was similar in the two waves, contrasting with record number of hospitalizations, daily deaths and health system collapse seen across the country in the second wave. Our findings suggest that the combination of the burden of severe cases and factors such as resource allocation and health disparities may have had an impact in the excess mortality found in many countries in the second wave.

## Introduction

The Coronavirus Disease 2019 (COVID-19) pandemic was the most severe public health crisis of the century. By July 2023, Brazil was the sixth country in number of confirmed cases in the world ranking, with more than 37 million cases [1], and second country in cumulative deaths, with more than 700,000 deaths registered [1]. Among critically ill patients with severe acute respiratory infection (SARI) and COVID-19-related acute respiratory distress syndrome (ARDS), mortality in the first wave was high [2], especially among patients who required mechanical ventilation [3].

As countries were recovering from the toll taken by the first COVID-19 wave, the rapid worldwide spread of SARS-CoV-2 contributed to the emergence of new genetic lineages, called "variants of concern" (VOCs) [4] which showed increased transmissibility [4–6], escaped immunity generated by vaccine or natural infection [7–9], and raised concerns about increased disease severity and mortality [10–12]. The VOCs led to the worsening of the world health scenario with subsequent waves of cases and deaths. In Brazil, the gamma variant, which originated in the Amazon region, was the driver of the second wave, which was evident by January 2021 after a period of 3 months of decreasing number of cases [1, 13]. In March 2021, the Brazilian government issued a warning about the presence of Gamma (P.1), Alpha (B.1.1.7) and Beta (B.1.351) variants in Brazil [14], with a proportion of the Gamma variant reaching approximately 90% of the SARS-CoV-2 lineages circulating in Brazil during this period [5].

During the second wave, the Brazilian health system collapsed with rapidly rising number of cases, record deaths [11, 15], ICU occupancy above 90% in several large cities [11, 16, 17], similarly to what was seen in other parts of the world [18–21]. The causes for excessive deaths, especially among the younger patients, were not clear. One initial hypothesis was that the Gamma variant could be more virulent, causing more serious illness, including among younger patients [22–25]. Other possible explanations were lack of access to vaccination, which was restricted to older and the most vulnerable populations [26, 27], strain on the health system due to increased transmissibility, limited access to intensive care beds and failure to implement public health interventions to contain community transmission [11, 28].

Currently, there are limited data comparing outcomes of critically ill patients across different waves of COVID-19 focusing on the emergence of VOCs [9, 12, 18]. More data are needed to clarify the interplay between different VOCs, disease severity and availability of health care

resources and its impact on mortality. Therefore, we designed a cohort study to compare characteristics, clinical management, and outcomes of critically ill patients hospitalized in the first and second waves of COVID-19 in a large academic hospital in Brazil. We hypothesized that in-hospital mortality would be greater in the second wave.

## Methods

### Study design and location

This was a retrospective cohort conducted at Hospital das Clínicas, a public hospital affiliated with the University of Sao Paulo Medical School, and the largest academic hospital in Brazil. The hospital is public, and patients are treated at no cost in accordance with the Brazilian universal health system. We compared admission characteristics and hospital outcomes of patients admitted to ICUs dedicated exclusively to the care of COVID-19 in the first and second waves of the pandemic. During the first wave, in 2020, there were 20 COVID-19 ICUs with a total of 300 beds, of which 206 beds were operating rooms or ward beds converted into ICU beds. At that time, our hospital complex was the most important referral center for COVID-19 patients from the metropolitan region of São Paulo comprising a population comparable to countries the size of Portugal. In 2021, during the second wave, referral of severe COVID-19 cases was more organized and less concentrated in our hospital, in which five ICUs remained dedicated to COVID-19, with 58 beds.

The Research Ethics Committee approved the study (number CAAE: 50340521.6.0000.0068). Requirement for informed consent form was waived because of the observational nature of the study.

### Study population

From March 30 until June 30, 2020 (first wave) and April 1$^{st}$ until July 31, 2021 (second wave), all consecutive patients admitted to the ICU with COVID-19 were screened. ICU admission was regulated by a team of physicians and nurses who received ICU admission requests from the emergency rooms and wards, and outside hospitals. Inclusion criteria were age $\geq$18 years and confirmed COVID-19 with Reverse Transcription Polymerase Chain Reaction (RT-PCR) or the rapid-antigen test or antibody (serology) test for IgM and IgG (more details in S1 Text). Exclusion criteria were palliative care in the first 24 hours after ICU admission, less than 24 hours of ICU stay, admission to the ICU after more than 7 days of invasive ventilatory support in another health institution or more than 14 days of hospitalization. We also excluded patients who were admitted to ICU due to other diagnosis and without symptoms of COVID-19, but who had SARS-CoV-2 RT-PCR positive on admission screening, defined as asymptomatic COVID-19.

### Patient care

Patient care was at the ICU teams' discretion. The hospital developed institutional protocols specifically for COVID-19 patients, since the beginning of the pandemic, including the best evidence that emerged related to the care of critically ill patients with COVID-19. The main treatment protocol changes between 2020 and 2021 were use of corticosteroid in hypoxemic patients, use of prophylactic doses of anticoagulants to patients without evidence of thromboembolism, use of antibiotics restricted to patients with suspected secondary bacterial infection [29] and reduced restrictions for the use of noninvasive ventilation (NIV) and high-flow nasal cannula (HFNC).

## Outcomes

The main outcome was 60-day hospital survival. The secondary outcomes were use of protective ventilation, rescue therapies for refractory hypoxemia, need for renal replacement therapy (RRT) and/ or vasopressors.

## Data collection

Patient information was collected from electronic medical records. Data were accessed from August 19, 2021, to November 12, 2021. We used an online case report form, managed on REDCap-Research Electronic Data Capture, an online platform, [30]. Confidentiality of the information was ensured by restricted access to the redcap database to the two principal investigators and use of deanonymized databases on export. We followed patients until hospital discharge or transfer to other health services.

We collected data related to ICU admission, management in the first 24 hours, and outcomes. These data included demographic information, comorbidities, duration of symptoms, laboratory tests, prognostic scores such as Simplified Acute Physiology Score III (SAPS 3) [31] and Sequential (Sepsis-related) Organ Failure Assessment (SOFA) [32], specific COVID-19 treatment, need for advanced life support therapies, and ventilator parameters. We defined protective ventilation as use of tidal volume $< 8$ mL/Kg and plateau pressure $< 30$ cmH$_2$O. We also collected ICU and hospital outcomes and clinical complications reported during the course of illness.

Information about vaccination against SARS-CoV-2 was collected from electronic medical records or telephone call to the patient's next of kin. In early 2021, vaccines approved for use in Brazil were those produced by Sinovac/Butantan Institute, AstraZeneca/Fiocruz, Pfizer/Wyeth and Janssen [27]. In São Paulo, vaccination started in mid-January 2021 for healthcare workers and was gradually expanded with priority for the most vulnerable population, the elderly and immunosuppressed. By April 2021, residents over 63 years-old had been offered their first shot and by July 2021, the cutoff was at 28 years-old [27].

Nasal swab and tracheal aspirate samples from patients admitted in the second wave were tested for detection of SARS-CoV-2 variants, using the QuantStudio™ 5 Real-Time PCR System (Applied Biosystem, Foster City, California, USA). Variant testing could not be performed for patients when samples were unavailable, either because RT-PCR was done in another health service, or because COVID-19 diagnosis was made using antigen or serologic tests (more details in S1 Text). Variant testing was not performed on samples from patients admitted in the first wave because at the time, there were no VOCs circulating in Brazil.

We report the results according to the recommendations the Strengthening The Reporting of Observational Studies in Epidemiology (STROBE) guidelines [33].

## Statistical analysis

Continuous variables were expressed as mean and standard deviation (SD) or median and interquartile range (IQR) as appropriate and compared using the independent samples Student's t-test or Mann-Whitney U test. Categorical variables were presented as absolute and relative frequencies and compared using the Chi-square test.

We plotted Kaplan–Meier curves to estimate 60-day survival in each of the pandemic waves. 60-day survival was defined as the time interval between ICU admission and patient death from any cause or hospital discharge. Patients discharged home or transferred to other health services were considered alive at the end of follow-up. In the unadjusted model, a log rank test was used to compare the survival of patients in the two pandemic waves. In addition, Cox proportional hazard models were used to compare the survival of patients in the two

pandemic waves, both without adjustments, and adjusting for potential confounders. We built the multivariable Cox model based on a conceptual causal diagram in the format of directed acyclic graph (DAG) [34], including the association between admission in the second wave and survival, adjusted by the most relevant variables. Variables in the DAG conceptual model were selected based on prior knowledge and are depicted in S1 Fig. A sensitivity analysis using an alternative DAG model not including vaccination was also performed (S2 Fig). We tested the proportional hazards assumption in Cox models with the Schoenfeld residuals method. The multivariable model did not include interaction terms or any other higher order terms.

Unadjusted and adjusted hazard ratios (HR) and 95% confidence intervals (95%CI) were used to measure the association between each variable and 60-day survival. All hypothesis tests are two-tailed and $p$-value $< 0.05$ was considered statistically significant.

The analyses were performed using the statistical software R (R Foundation for Statistical Computing Platform, version 4.2.1) [35–38].

## Results

### Study population

We screened 1,955 patients admitted to the ICU during the study periods, and included 1,583 patients, of whom 1,315 were admitted in the first and 268 were admitted in the second wave (Fig 1).

Follow-up was complete for all patients until hospital discharge, death in the hospital or transfer to other health services. In 2020, 104 (8%) patients were transferred to a long-term care facility, and in 2021 only 4 (1.5%) patients were transferred.

### Baseline characteristics

The baseline characteristics of the subjects, stratified by year of admission are shown in Table 1. In the first wave, patients were older, 742 (56%) were ≥60 years old, than in the second wave, when 114 (43%) were ≥60 years old. Male sex was predominant in both periods, 795 (61%) in the first wave and 165 (62%) in the second wave. At ICU admission, patients in the first wave were more likely to be receiving mechanical ventilation (61% vs 45%, $p<0.001$) and vasopressors (39% vs 30%, $p = 0.005$), had more comorbidities, and the mean SAPS3 (64 ± 16 vs 56 ± 13, $p<0.001$) and the median SOFA 7 [3–10] vs 4 [2–7], $p<0.001$) were higher. Laboratory findings are shown in S1 Table.

### Vaccination and variants of concern

In the second wave, nearly a third of patients were vaccinated with at least one dose of the COVID-19 vaccine. 67 (25%) patients in the second wave had samples available for SARS-CoV-2 mutations testing, 94% of whom were found to be infected with the Gamma variant (Table 1).

### Ventilatory management in the first 24 hours after ICU admission

More patients required mechanical ventilation in the first 24 hours of their ICU stay in the first wave compared with the second, 878 (67%) vs 148 (55%), $p<0.001$, as shown in Table 2. The use of protective ventilation was common and similar in both waves (82% in the first wave vs 88% in the second wave, $p = 0.08$).

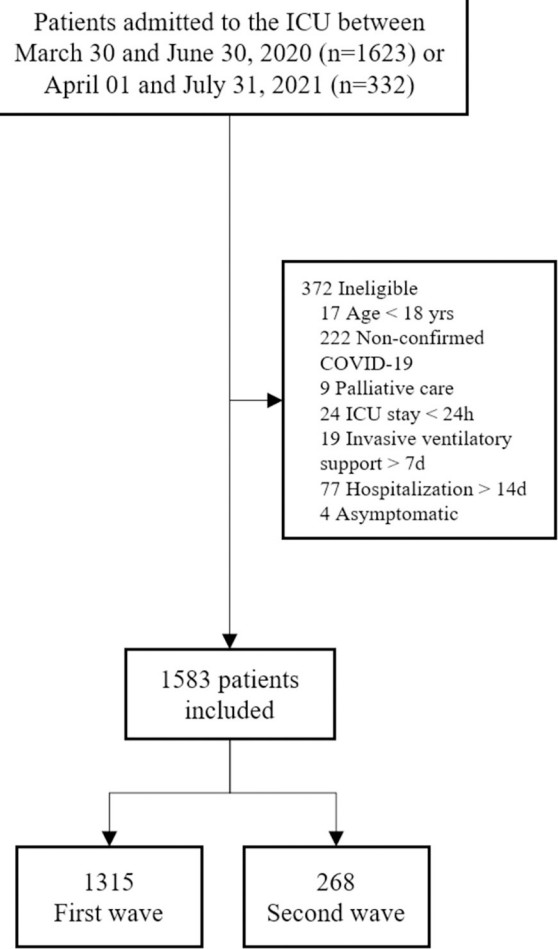

**Fig 1. Study participant flow chart.** Flow of potentially eligible participants in the study, and final numbers included and analyzed. ICU: Intensive Care Unit; COVID-19: Coronavirus Disease 2019.

## Clinical management of patients in the first 24 hours after ICU admission

Differences in the clinical management of patients in the first 24 hours after ICU admission between waves are shown in S2 Table. In the first wave, the percentage of patients under sedation was higher than in the second wave (63% vs 55%, $p < 0.001$), but use of neuromuscular blockade was more common on the second wave (27% in the first wave vs 39% in the second wave, $p < 0.001$). In the second wave, the use of antibiotics was less common (69% vs 42%, $p < 0.001$), and therapeutic anticoagulation was more frequently used in the second wave (9% vs 19%, $p < 0.001$). Systemic corticosteroids were used for 98% of patients in the second wave, and only for 25% of patients in the first wave.

## ICU and hospital outcomes

Relevant clinical outcomes are shown in Table 3. The median ICU stay was 11 [6–19] days in the first wave and 10 (7–19) in the second wave($p = 0.445$). Hospital stay was longer, 21 days [14–32], on the second wave compared to the first wave, 17 days [11–27], $p < 0.001$. More patients in first wave required mechanical ventilation (80% vs 70%, $p < 0.001$), while the use of NIV and HFNC were more frequent in the second wave, (22% vs 46%, $p < 0.001$) and (10% vs 41%, $p < 0.001$), respectively. The duration of mechanical ventilation was similar for the two

**Table 1. Baseline characteristics at ICU admission.**

| | First wave (n = 1315) | Second wave (n = 268) | *p* value |
|---|---|---|---|
| **Characteristics** | | | |
| Age (y), mean (SD) | 61 ± 15 | 56 ± 14 | <0.001 |
| **Age groups (y), n (%)** | | | <0.001 |
| 18–40 | 138 (11) | 46 (17) | |
| 41–60 | 435 (33) | 108 (40) | |
| >60 | 742 (56) | 114 (43) | |
| Female sex, n (%) | 520 (39) | 103 (38) | 0.787 |
| BMI | 28 ± 7 | 30 ± 6 | <0.001 |
| SAPS 3 | 64 ± 16 | 56 ± 13 | <0.001 |
| SOFA, median [IQR] | 7 [3 – 10] | 4 [2 – 7] | <0.001 |
| Duration of symptoms (d), median [IQR] | 9 [6 – 12] | 12 [9 – 14] | <0.001 |
| Vasopressors, n (%) | 515 (39) | 80 (30) | 0.005 |
| Invasive mechanical ventilation, n (%) | 809 (61.5) | 122 (45.5) | <0.001 |
| Corticosteroids, n (%) | 146 (11) | 207 (77) | <0.001 |
| Vaccination, n (%) | 0 (0.0) | 77 (29) | - |
| 1 dose, n (%) | 0 (0.0) | 52 (20) | |
| 2 doses, n (%) | 0 (0.0) | 25 (9) | |
| SARS-CoV-2 variant screening, n (%) | 0 (0.0) | 67 (25) | - |
| Gamma (P1), n (%) | 0 (0.0) | 63 (94) | |
| Alpha (B.1.1.7), n (%) | 0 (0.0) | 1 (1.5) | |
| Indeterminate, n (%) | 0 (0.0) | 3 (4.5) | |
| **Race[a], n (%)** | | | <0.001 |
| White | 795 (60) | 201 (75) | |
| Black | 94 (7) | 8 (3) | |
| Mix-ethnicity (Pardo) | 364 (28) | 47 (18) | |
| Asian | 15 (1) | 3 (1) | |
| Not informed | 47 (4) | 9 (3) | |
| **Comorbidities, n (%)** | | | |
| Asthma | 36 (3) | 13 (5) | 0.104 |
| Cancer | 129 (10) | 10 (4) | 0.002 |
| Cardiovascular disease | 194 (15) | 54 (20) | 0.034 |
| Cerebrovascular disease | 54 (4) | 5 (2) | 0.112 |
| Chronic kidney disease | 129 (10) | 7 (3) | <0.001 |
| Chronic pulmonary disease | 77 (6) | 17 (6) | 0.868 |
| Diabetes | 501 (38) | 79 (29) | 0.009 |
| Hypertension | 751 (57) | 122 (46) | 0.001 |
| Obesity | 423 (35) | 136 (51) | <0.001 |
| HIV/AIDS | 15 (1) | 1 (0.4) | 0.418 |

BMI: body mass index, kg/m2; IQR: interquartile range; SAPS 3: Simplified acute Physiology Score 3; SOFA: Sepsis-related Organ Failure Assessment. Data are presented as mean and standard deviation, unless otherwise stated; comparisons were made with t test, Mann–Whitney U test or Chi-square test as appropriate. Missing year 2020: BMI for 106 (8%) patients; SAPS3, missing for 1 patient. Missing year 2021: BMI for 1 patient, SARS-CoV-2 variant screening for 201 (75%) patients in 2021.

[a] The categories represent the Brazilian official race categories.

waves, 10 [6–17] vs 10 [6–19], *p* = 0.458. Prone position was more commonly used in the second wave (38% vs 57%, *p*<0.001), while vasopressors and RRT were required more often in the first wave (74% vs 60%, *p*<0.001).

**Table 2. Ventilatory management on the first 24 h after ICU admission.**

| Management | First wave (n = 878) | Second wave (n = 148) | p value |
|---|---|---|---|
| Tidal volume (mL/Kg ibw), mean (SD) | 6.55 ± 1.3 | 6.17 ± 1.2 | 0.001 |
| Respiratory Rate, median [IQR] | 30 [26 - 35] | 30 [25 - 31] | 0.015 |
| Minute volume, mean (SD) | 12 ± 3.7 | 11 ± 2.7 | <0.001 |
| FiO2 (%), median [IQR] | 50 [40 - 60] | 50 [40 - 70] | 0.178 |
| PEEP (cmH2O), median [IQR] | 10 [8 - 12] | 10 [8 - 14] | <0.001 |
| Plateau pressure (cmH2O), mean (SD) | 22.6 ± 4.7 | 23.9 ± 4.3 | 0.001 |
| Driving pressure (cmH2O), mean (SD) | 12.6 ± 4 | 12.8 ± 3 | 0.686 |
| Compliance (mLcmH2O$^{-1}$), median [IQR] | 32 [24 - 41] | 30 [24 - 37] | 0.070 |
| Compliance (mLcmH2O$^{-1}$.Kg$^{-1}$ibw), median [IQR] | 0.52 [0.41–0.65] | 0.48 [0.39–0.62] | 0.022 |
| PaO2/FIO2 (%), mean (SD) | 168 ± 70 | 164 ± 72 | 0.521 |
| Arterial pH, median [IQR] | 7.36 [7.30 - 7.42] | 7.36 [7.29 - 7.42] | 0.986 |
| Arterial PaCO2 (mmHg), median [IQR] | 42.3 [37.9 - 48.3] | 47.00 [39.0 - 56.1] | <0.001 |
| Arterial O2 saturation (%), median [IQR] | 93 [91 - 96] | 94 [91 - 96] | 0.687 |
| **Ventilation Mode, n (%)** | | | <0.001 |
| Volume-controlled ventilation | 503 (57) | 94 (64) | |
| Pressure-controlled ventilation | 167 (19) | 28 (19) | |
| Pressure support ventilation | 199 (23) | 15 (10) | |
| Other | 9 (1) | 11 (7) | |
| **Rescue therapy for hypoxemia, n (%)** | | | |
| Prone position | 145 (16) | 61 (41) | <0.001 |
| PEEP titration | 103 (12) | 57 (38) | <0.001 |
| Recruitment maneuvers | 14 (1.6) | 12 (8.1) | <0.001 |
| Extracorporeal membrane oxygenation | 1 (0.1) | 2 (1.4) | 0.079 |
| Inhaled nitric oxide | 1 (0.1) | 0 (0.0) | - |
| **Protective ventilation, n (%)** | 634 (82) | 130 (88) | 0.081 |

SD: standard deviation; IQR: interquartile range; O2: oxygen; PaCO2: arterial partial pressure of carbon dioxide; FIO2: inspired fraction of oxygen; PEEP: positive end-expiratory pressure; ibw: ideal body weight; PaO2/ FIO2: partial pressure of arterial oxygen and fraction of inspired oxygen ratio. Data are n. (%), unless otherwise stated; comparisons were made with t-test, Mann–Whitney U test or Chi-square test as appropriate.

Missing year 2020: Tidal volume (mL/Kg ibw) for 15 (1.7%) patients; PaO2/FIO2 was missing for 4 (0.5%) patients; plateau pressure (cmH2O) and driving pressure (cmH2O) were missing for 95 (11%) patients; PaCO2 for 58 (7%) patients; arterial pH for 52 (6%) patients and arterial O2 saturation for 61 (7%) patients. Missing year 2021: none.

At the end of 60-days follow-up, 645 (49%) patients died in the first wave and 97 (36%) patients died in the second wave. After 60 days, another 13 (1%) patients in the first wave and 5 (2%) in the second wave died in the hospital.

## Association of baseline characteristics with survival

Admission in the second wave was associated with higher survival at 60 days in the unadjusted analysis (logrank, p<0.001 and HR 0.61, 95%CI 0.49–0.76), as shown in Fig 2A, and in S3 Table. After adjusting for age, SAPS 3 and vaccination, according to our conceptual causal diagram, admission in the second wave was no longer associated with survival (aHR 0.85, 95%CI 0.65–1.12). In this multivariable model, the only variables independently associated with 60-day hospital survival were age (aHR 1.02, 95%CI 1.02–1.03) and SAPS 3 (aHR 1.03, 95%CI 1.03–1.04), shown in in Fig 2B, and in S3 Table. In the sensitivity analysis excluding vaccination from the DAG model, we found similar results, with admission not associated with survival (aHR 0.89, 95%CI 0.72–1.12), as shown in S4 Table.

**Table 3. Clinical outcomes.**

| Outcomes | First wave (n = 1315) | Second wave (n = 268) | p value |
|---|---|---|---|
| ICU length of stay, median [IQR], d | 11 [6 - 19] | 10 [7- 19] | 0.445 |
| Hospital length of stay, median [IQR], d | 17 [11 - 27] | 21 [14 - 32] | <0.001 |
| Invasive mechanical ventilation, n (%) | 1051 (80) | 187 (70) | <0.001 |
| Duration of mechanical ventilation, median [IQR], d | 10 [6 - 17] | 10 [6 - 19] | 0.458 |
| Reintubation, n (%) | 206 (15.7) | 17 (6.3) | <0.001 |
| Prone positioning, n (%) | 401 (38) | 107 (57) | <0.001 |
| Noninvasive ventilation[a], n (%) | 291 (22) | 123 (46) | <0.001 |
| High-flow nasal cannula[a], n (%) | 128 (10) | 109 (41) | <0.001 |
| Extracorporeal membrane oxygenation, n (%) | 6 (0.5) | 5 (1.9) | 0.033 |
| Vasopressors, n (%) | 976 (74) | 162 (60) | <0.001 |
| Renal replacement therapy, n (%) | 481 (37) | 58 (22) | <0.001 |
| Tracheostomy, n (%) | 160 (12) | 32 (12) | 0.999 |
| Delirium, n (%) | 430 (33) | 28 (10) | <0.001 |
| Ventilator-associated pneumonia, n (%) | 358 (27) | 79 (29) | 0.479 |
| Thromboembolic event, n (%) | 246 (19) | 81 (30) | <0.001 |
| Cardiac arrhythmia, n (%) | 223 (17) | 43 (16) | 0.769 |
| Treatment withhold or withdraw during hospital stay, n (%) | 249 (19) | 22 (8) | <0.001 |
| Mortality at 28 days, n (%) | 580 (44) | 87 (32) | <0.001 |
| Mortality at 60 days, n (%) | 645 (49) | 97 (36) | <0.001 |
| **ICU outcome, n (%)** | | | <0.001 |
| Discharged home | 22 (1.7) | 10 (3.7) | |
| Discharged to the ward | 641 (49) | 162 (60) | |
| Transferred to another ICU | 50 (3.8) | 6 (2.2) | |
| Transfer to long-term care facility | 1 (0.1) | 1 (0.4) | |
| Death | 601 (46) | 89 (33) | |
| **Hospital outcome, n (%)** | | | <0.001 |
| Discharged home | 553 (42) | 162 (60) | |
| Transfer to long-term care facility | 104 (8) | 4 (1.5) | |
| Death | 658 (50) | 102 (38) | |

ICU: intensive care unit; IQR: interquartile range; Data are n. (%), unless otherwise stated; comparisons were made with t test, Mann–Whitney U tests or chi-square test as appropriate.

[a] To avoid intubation or prior to intubation.

## Discussion

In this retrospective cohort, we compare characteristics and outcomes of 1,583 patients with COVID-19 admitted to the ICUs of a referral center for COVID-19 in Brazil in the first two waves of the pandemic. We found that 60-day survival was higher in the second wave compared to the first wave. However, in the multivariable analysis, adjusting by SAPS3, age and vaccination, admission to the ICU in the second wave was no longer associated with mortality. Patients admitted to the ICU in the second wave were younger, had better prognostic scores, and less need for vasopressors and RRT. Invasive mechanical ventilation was needed by most patients in both waves, 80% in the first and 70% in the second wave, and noninvasive ventilatory support and prone increased substantially in the second wave. Duration of mechanical ventilation and ICU length of stay were similar in the two waves, while hospital length of stay was longer in the second wave.

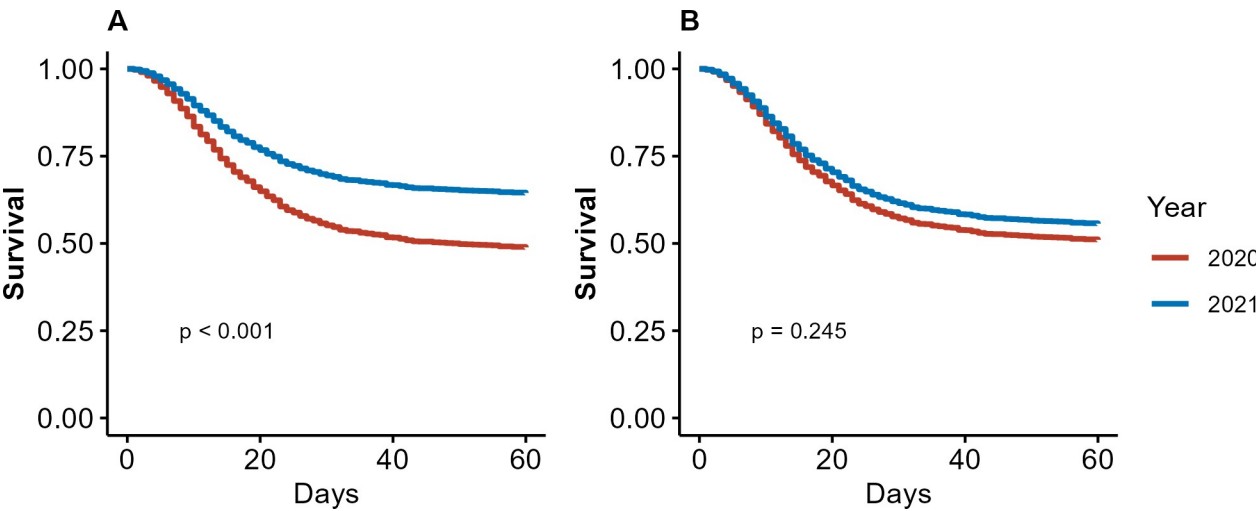

**Fig 2. Hospital 60-day survival according to year of admission.** Solid red line represents survival of patients who were admitted in 2020 (first wave) and solid blue line represents survival of patients who were admitted in 2021 (second wave). (A) Unadjusted survival, logrank $p < 0.001$ and HR 0.61, 95%CI 0.50–0.76. (B) Survival adjusted by age, SAPS 3 and vacination (aHR 0.85, 95%CI 0.65–1.12). The $p$ values were obtained with Cox proportional hazards models.

Survival at 60 days in our hospital was higher in the second wave than in the first wave, contrary to our study hypothesis, but in line with what was observed in studies from other countries [39–43]. Possible reasons for this difference include the admission of younger and less severe patients, earlier hospitalization, more organized and efficient ICU care, with better trained health care professionals, and more knowledge about the disease with better treatment protocols in the second wave. The impact of younger age and less severe disease at ICU admission on survival is supported by the finding that survival was similar in both waves when adjusted for age and SAPS3.

Our results contrast with the catastrophic outcomes seen in Brazil during the second wave, one the most affected countries in the world. By February 2021, the rapid spread of the Gamma variant was associated with a steep increase in number of cases [15], accompanied by a rise in the number of cases of SARI [44]. This combination of factors led to record hospitalizations and deaths [15] with signs of health system collapse [11, 15, 16], resulting in worse outcomes and potentially preventable deaths [45, 46].

Access to ICU beds and availability of resources are very discrepant across Brazil, and such disparities increased during the pandemic [45]. Availability of ventilators was up to 5 times greater in southern and southeastern capitals compared to northern and northeastern capitals [45], and age-standardized mortality rates varied by up to 5 times between the northern and southern states in the second wave [44, 47]. Our hospital is university-affiliated, located in the richest state of the country, with capability to increase ICU capacity and resources according to demand, and a centralized referral system which limited the number of patients admitted each day, in a striking contrast with frontline hospitals in the Amazon region, for example.

In Brazil and other countries, the second wave had an increased proportion of younger and previously healthy patients who required ICU care [15, 39–41]. This reduction in the average age could be related to the fact that vaccination coverage started in January 2021 in Brazil and was associated with a reduction in hospitalizations and deaths in adults older than 70 years [26], but was not available to younger populations at that time. Another possible explanation to the high burden of cases among younger people in Brazil was a greater exposure to the virus

during the second wave, when fewer restrictive measures were in effect, adherence to the use of masks decreased, and part of the economically active population had returned to work [28].

Only 25 (9%) of patients of the second wave had completed two doses of vaccination when they were admitted to the ICU. This low number reflects the vaccination calendar in Brazil, since vaccination started in January 2021 and was scaled by age, starting with very old citizens [27]. As a result, despite documented protection against COVID-19 [26], vaccination probably had very little impact for patients of the second wave included in our study.

Patients in the second wave were less likely to need vasopressors and RRT during their ICU stay than patients in the first wave, as noted in other studies [39, 40, 48]. These results support the finding that better clinical outcomes in the second wave could be associated with younger age and less severe disease at ICU admission.

During the second wave, we found that 94% of the samples analyzed detected the Gamma variant, in accordance to epidemiological findings in Brazil during this period [5]. The Gamma variant spread throughout Brazil and other countries and was found to be more transmissible [4–6], having a higher risk of hospitalization [44, 49]. While some studies found that it caused more severe disease and increased the risk of death [44, 46, 50], others did not see an increased mortality [48, 51, 52]. It is possible that the elevated daily death counts observed during the second wave in Brazil was due to a combination of increased risk of development of SARI [44] and lack or delayed access to healthcare and strain on the healthcare system. Future comparisons with other cohorts of patient may bring addition light to the role of VOCs on severe COVID-19.

Management of patients in the second wave was influenced by accumulated knowledge about the efficacy of several therapeutic options. Importantly, less use of antibiotics [53], and more use of corticosteroids were informed by randomized controlled trials [54, 55]. In our cohort, only 25% of the patients received corticosteroid in the first wave, while 98% received it in the second wave, which may have had an impact in survival.

The use of noninvasive ventilatory support increased during the second wave, likely due to greater availability, more experience, and less concern about and environmental contamination [56]. Use of noninvasive support may have contributed to avoiding intubations in the second wave, in which we found 10% less use of invasive mechanical ventilation compared to the first wave. Currently, there is evidence that noninvasive ventilatory support is effective in decreasing intubation rates, but not mortality, compared to conventional oxygen in COVID-19 [57, 58].

Among intubated patients, we found that the more than 80% of patients received protective ventilation in both waves, which our group previously found to be associated with increased survival in COVID-19 [2]. Rescue therapies, including recruiting maneuvers, PEEP titration, and extracorporeal membrane oxygenation were more often used in the second wave. The same observation was seen with prone position, which was used in 57% of the patients in the second wave, compared to 38% in the first wave. Prone position was increasingly used during the COVID-19 pandemic, probably based on better training of health professionals, greater knowledge in the management of patients with COVID-19 pneumonia, and previous evidence of benefit of prone in patients with moderate to severe non-COIVD-19 ARDS [59].

Our study has several limitations. Being a retrospective observational study, data were collected from the electronic medical record, which could have incomplete or inaccurate data. It is a single-center study, and our findings may not be generalizable to other hospitals in Brazil and other low-and-middle-income countries. However, it is our belief that this does not have a significant impact on the inferential conclusions related to the comparison between COVID-19 waves. Only 25% of respiratory samples from patients in the second wave were available for variant testing, because many patients were diagnosed in other health services before being

transferred to our hospital or diagnosed with antigen tests, limiting our power to estimate the association of variants of concern with clinical outcomes. We compared the first two waves of COVID-19 in Brazil, which happened in different contexts of hospital organization. During the first wave, our hospital had five times more ICU beds compared to the second wave, reflecting greater strain. Our study was unable to measure the impact of updated treatment protocols, better structuring of ICUs and training of health care professionals in our hospital that occurred in the second wave. We included vaccination to our DAG conceptual model, but since vaccination was not available for patients in the first wave, or for younger patients in the second wave, its inclusion might bias our results and impact the interpretation of the results. In order to address this issue, we performed a sensitivity analysis removing vaccination from the model and found similar results. The sample of patients admitted in the second wave was smaller, which limited our statistical power. Finally, since our study only included ICU patients, we could not measure the impact of the Gamma variant on increased risk of development of SARI, which in turn may have been responsible for overall increased risk of death in the second wave. Such limitations may impact the generalizability of our results and the interpretation of the results, particularly the significance of lower mortality observed in our hospital during the second wave, when mortality was higher in official country-wide data.

Our study also has strengths. The hospital covers a large geographical area, with an estimated population of 23 million; we screened all consecutive patients with confirmed COVID-19 admitted to the participating ICUs, minimizing selection bias; we included a large sample size; we had complete follow-up of patients until hospital discharge or transfer to another health service; we had few missing data; and, we assessed objective, hard outcomes. In addition, our hospital followed evidence-based clinical management protocols, which were reviewed as evidence became available. These results corroborate the importance of implementing up to date clinical practice guidelines during health emergencies to provide rational allocation of resources and improve patient outcomes.

## Conclusion

In this cohort study conducted across multiple ICUs of the largest academic hospital in Brazil, we observed that patients admitted to the ICU in the second wave of the COVID-19 pandemic were younger and had better prognostic scores compared to those in the first wave. They also required fewer advanced life support therapies and had higher survival rates. Survival rates adjusted for age and severity score were found to be similar between waves. These results contrast with the country-level outcomes and underscore the impact of disease severity at ICU admission, as well as the availability and rational allocation of healthcare resources, health-system strain, and health disparities on the outcome of the pandemic. As the world starts to recover from the impacts of the COVID-19 pandemic, other epidemics continue to threaten populations and healthcare systems. Our results shed light on the interplay between infectious agents' virulence, patient vulnerabilities and resource availability and can inform preparedness strategies to respond to epidemics and provide equitable care to all.

## Supporting information

**S1 Text. Supplementary methods.**
(DOCX)

**S1 Fig. Causal diagram in the format of directed acyclic graph (DAG).** SAPS 3: Simplified acute Physiology Score 3. This conceptual model shows clinically relevant variables associated with survival. Arrows indicate a presumed direct causal effect of one variable on another variable.

Admission in the second wave is the main predictor, shown in green; variables associated with the outcome, but not associated with the main predictor, are shown in blue; variables associated with both the outcome and the main predictor are shown in red (arrows indicate a suspected direct causal effect of that variable on both the main predictor and the outcome). A multivariate analysis for estimating the direct effect of admission in the second wave on survival should be adjusted for potential confounders, identified in the model as age, SAPS3 and vaccination. (TIF)

**S2 Fig. Alternative causal diagram in the format of directed acyclic graph (DAG) used for a sensitivity analysis.** SAPS 3: Simplified acute Physiology Score 3. This conceptual model, used as a sensitivity analysis, shows clinically relevant variables associated with survival, not including vaccination. Arrows indicate a presumed direct causal effect of one variable on another variable. Admission in the second wave is the main predictor, shown in green; variables associated with the outcome, but not associated with the main predictor, are shown in blue; variables associated with both the outcome and the main predictor are shown in red (arrows indicate a suspected direct causal effect of that variable on both the main predictor and the outcome). A multivariate analysis for estimating the direct effect of admission in the second wave on survival should be adjusted for potential confounders, identified in the model as age and SAPS3. (TIF)

**S1 Table. Laboratory tests at ICU admission.** Data are presented as median [IQR]: interquartile range; comparisons were made with Mann-Whitney test. Missing year 2020: Arterial lactate for 399 (30%) patients; D-dimer for 316 (24%) patients; Arterial pH for 151 (12%) patients; C-reactive protein for 202 (15%) patients. Missing year 2021: Arterial lactate for 65 (24%) patients; D-dimer for 12 (5%) patients; Arterial pH for 45 (17%) patients; C-reactive protein for 10 (4%) patients. (DOCX)

**S2 Table. Patient management on the first 24 h after ICU admission.** Definition of abbreviations: O2: oxygen; RASS: Richmond Agitation-Sedation Scale; COVID-19: Coronavirus Disease 2019. Data are n. (%); comparisons were made with the chi-square test. (DOCX)

**S3 Table. Association between admission in the second wave, other relevant covariates, and 60-day survival.** HR: hazard ratio; aHR: adjusted hazard ratio; 95%CI: 95% confidence interval; SAPS 3: Simplified acute Physiology Score 3; HR, aHR obtained with univariate and multivariate Cox models, respectively, and 95%CI and p values obtained in each model. SAPS 3 was missing for 1 patient. (DOCX)

**S4 Table. Sensitivity analysis—association between admission in the second wave, other relevant covariates, and 60-day survival.** HR: hazard ratio; aHR: adjusted hazard ratio; 95% CI: 95% confidence interval; SAPS 3: Simplified acute Physiology Score 3; HR, aHR obtained with univariate and multivariate Cox models, respectively, and 95%CI and p values obtained in each model. SAPS 3 was missing for 1 patient. (DOCX)

## Acknowledgments

We would like to thank the Hospital das Clinicas COVID-19 crisis committee, healthcare workers and staff for their important work at our hospital during the COVID-19 pandemic.

## Author Contributions

**Conceptualization:** Daniela Helena Machado Freitas, Carlos Roberto Ribeiro Carvalho, Juliana Carvalho Ferreira.

**Data curation:** Daniela Helena Machado Freitas, Juliana Carvalho Ferreira.

**Formal analysis:** Daniela Helena Machado Freitas, Eduardo Leite Vieira Costa, Juliana Carvalho Ferreira.

**Methodology:** Daniela Helena Machado Freitas, Eduardo Leite Vieira Costa, Pâmela Santos Andrade, Flávia Cristina Silva Sales, Ester Cerdeira Sabino, Carlos Roberto Ribeiro Carvalho, Juliana Carvalho Ferreira.

**Project administration:** Daniela Helena Machado Freitas, Carlos Roberto Ribeiro Carvalho.

**Supervision:** Juliana Carvalho Ferreira.

**Writing – original draft:** Daniela Helena Machado Freitas.

**Writing – review & editing:** Daniela Helena Machado Freitas, Eduardo Leite Vieira Costa, Natalia Alcantara Zimmermann, Larissa Santos Oliveira Gois, Mirella Vittig Alves Anjos, Felipe Gallego Lima, Pâmela Santos Andrade, Daniel Joelsons, Yeh-Li Ho, Flávia Cristina Silva Sales, Ester Cerdeira Sabino, Carlos Roberto Ribeiro Carvalho, Juliana Carvalho Ferreira.

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
