## [Decision Letter · Decision Letter 0]

21 Nov 2023

PONE-D-23-21314TEMPORAL TRENDS OF SEVERITY AND OUTCOMES OF CRITICALLY ILL PATIENTS WITH COVID-19 AFTER THE EMERGENCE OF VARIANTS OF CONCERN: A COMPARISON OF TWO COHORTSPLOS ONE

Dear Dr. Freitas,

Thank you for submitting your manuscript to PLOS ONE. After careful consideration, we feel that it has merit but does not fully meet PLOS ONE’s publication criteria as it currently stands. Therefore, we invite you to submit a revised version of the manuscript that addresses the points raised during the review process.

We look forward to receiving your revised manuscript.

Kind regards,

Veranyuy Ngah, MSc

Academic Editor

PLOS ONE

Journal Requirements:

"Dr. Ferreira reports personal fees from Medtronic, outside the submitted work; Dr. Costa reports personal fees from Timpel, personal fees from Magnamed, outside the submitted work; Dr. Ho reports personal fees from Pan‐American Health Organization, outside the submitted work. The other authors have no conflict of interest to disclose."

Reviewers' comments:

Reviewer's Responses to Questions

**Comments to the Author**

1. Is the manuscript technically sound, and do the data support the conclusions?

Reviewer #1: Yes

Reviewer #2: Yes

2. Has the statistical analysis been performed appropriately and rigorously? 

Reviewer #1: Yes

Reviewer #2: Yes

3. Have the authors made all data underlying the findings in their manuscript fully available?

Reviewer #1: Yes

Reviewer #2: Yes

4. Is the manuscript presented in an intelligible fashion and written in standard English?

Reviewer #1: Yes

Reviewer #2: Yes

5. Review Comments to the Author

Reviewer #1: Summary

Thank you for the opportunity to provide a peer review for this interesting article on “Temporal trends of severity and outcomes of critically ill patients with COVID-19 after the emergence of variants of concern”. The article compares critically ill patients' outcomes between two cohorts in the first two waves of COVID-19, comprising a sample of 1583 patients (1315 in the first and 268 in the second wave) from the largest academic hospital in Brazil. Results indicated that in the second wave, admitted COVID-19 ICU patients were younger and had better prognostic scores with higher survival rates compared to those in the first wave. However, the survival rates in the model adjusted for age and severity score were similar between waves; therefore, the authors conclude that factors such as resource allocation and health disparities could have impacted the excess mortality in many countries in the second wave.

Overall Impression

The study addresses an important research question and presents valuable insights into the outcomes of critically ill COVID-19 patients during the first two waves of the pandemic in Brazil. As is typically the case with the peer review process, I believe there are areas where the manuscript could be strengthened. With some revisions and additional contextual information, this manuscript has the potential to make a significant contribution to the field of pandemic healthcare management. To forecast the essence of my review, these recommendations can be interpreted as minor-to-major in nature. Although the effort required to address these recommendations is minimal, the flow-on effects on manuscript quality will be excellent. Please feel free to address these points in your revision, and don't hesitate to reach out if you need further clarification or assistance. Thank you for your contribution to the scientific community.

Specific Comments:

Abstract

Consider providing a brief context for readers unfamiliar with the Brazilian healthcare system and the significance of studying outcomes across different waves of the pandemic.

Introduction

Provide references to the claim made on P5 L101.

Methods

I like how the authors provided a very good description of the study population, highlighting the selection criteria. However, additional information is needed to clarify the specific criteria that were used to define critically ill patients. Detailing parameters like respiratory rate, oxygen saturation, and comorbidities would be helpful for readers to gauge the severity of the cases.

Kindly add references to the Patient Care subsection P7 L143.

On P7 L143, the authors mention the main changes in treatment protocol between 2020 and 2021. Can they also comment on the use of anticoagulants at prophylactic doses to prevent thromboembolism and the nonuse of antibiotics in patients without suspected bacterial infection as reported by Falavigna et al (https://www.jbmede.com.br/index.php/jbme/article/download/69/50).

In the Statistical Analysis subsection P8 L180, several essential details are missing that are crucial for a comprehensive understanding of the statistical analysis. The description mentions that Kaplan-Meier curves were plotted for each of the pandemic waves. It's important to clarify if these curves were plotted separately for each group (first wave vs. second wave) or if there was a comparison between the curves. If a comparison was made, the results of the log-rank test, which is commonly used to compare Kaplan-Meier curves between groups, should be reported.

Kindly explain the rationale for choosing the variables to be included in the multivariable Cox model-building process. There is no mention of how assumptions such as the proportional hazards assumption in Cox models were checked. It's vital to confirm that these assumptions hold for the validity of the analysis.

Information on how the goodness-of-fit of the Cox proportional hazards model was assessed is missing. Techniques like likelihood ratio tests, Akaike Information Criterion (AIC), or Bayesian Information Criterion (BIC) are commonly used to assess model fit.

Details about specific packages or functions within R used for the analyses are important for transparency and reproducibility.

Results

I liked the clear presentation of the patients’ characteristics from both waves in Table 1, including demographic information, comorbidities, and other relevant factors summarizing the key differences. This will provide a comprehensive understanding of the patient population studied.

In the Baseline Characteristics subsection P10 L212-219, the authors need to provide specific numerical data from Table 1 to support their statements about the subjects. Instead of using phrases like "male sex was predominant" or "more likely," the authors should directly quote the figures from the table to enhance the precision and clarity of the description. For instance:

Instead of saying "male sex was predominant in both periods," specify the exact percentages of male patients in both the first and second waves.

Instead of saying "56% were ≥60 years old" and "43% were ≥60 years old," provide the actual numbers of patients in each age group (percentage in brackets) for both waves.

Instead of saying, "Patients in the first wave were more likely to be receiving mechanical ventilation and vasopressors," provide the specific percentages or counts of patients receiving mechanical ventilation and vasopressors in each wave.

Provide the actual numerical values for mean SAPS3 and SOFA scores in both waves to convey a more precise comparison.

By directly citing the figures from Table 1, readers can have a clear and accurate understanding of the baseline characteristics of the subjects in each wave without having to move back and forth to Table 1.

The same applies to the ventilatory management in the first 24 hours after ICU admission (P12 L225-232) and management in the first 24 hours after ICU admission (P13 L240-247) subsections. See Lalla et al as an example (https://pubmed.ncbi.nlm.nih.gov/35359698/).

On P13 L240, the heading seems a bit vague; the authors can consider changing it to Clinical Management of Patients in the First 24 Hours After ICU Admission.

In the ICU and hospital outcomes subsection P14 L248-257, similar comments to the "Baseline Characteristics" subsection apply. In addition, include p-values or other indicators of statistical significance to assess whether the observed differences are likely due to chance or represent true disparities between the waves.

On P16 L274, the authors mention the multivariable model with vaccination. Adjusting for a variable that is measured in one group but not in another can introduce bias and affect the validity of the results. In this case, adjusting for vaccination as a variable in the Cox model only for the second wave raises some concerns:

(1) Selection Bias: Since vaccination status was only measured in the second wave, it means that the two groups (first wave and second wave) are not directly comparable concerning this variable. The absence of vaccination data in the first wave introduces selection bias, potentially leading to distorted results.

(2) Limitation of generalizability: The results might not be generalizable to populations where vaccination status is differentially available or measured between waves. This limitation could affect the study's broader applicability and relevance.

In the absence of uniform vaccination data, the study could acknowledge this limitation explicitly. Authors should discuss the implications of this limitation on the interpretation of results and potential biases introduced by the unequal measurement of vaccination status between waves.

Discussion

The authors fully explored potential reasons behind the observed differences in patient outcomes between the two waves. They discussed factors such as healthcare infrastructure and public awareness. Limitations of the study were mentioned; however, the authors may want to discuss further how these limitations might have influenced the results and interpretations.

Conclusion

Consider discussing the implications of the study findings for clinical practice. How can the insights gained from this study inform future pandemic preparedness and response strategies?

Reviewer #2: Overall impression and relevance

The study presents a relevant issue concerning variants of concerns and continuous outbreaks of the COVID-19. It is professionally written, and the methods used are detailed and comprehensive. However, from the topic and aim presented in the introduction, one would expect to see detailed analysis of how the different variants of concern influence survival/mortality in both the first wave and the second wave. But the study presents a comparison of survival/mortality of patients based on admission clinical characteristics during the first 24 hours as its primary outcome and assessing the difference between other clinical outcomes between the 1st and 2nd wave as secondary outcomes. The authors should consider changing the topic and aim to align with the methods and results.

Background

1 The statement on pg. 5, L 102-103 requires backing with references.

Methods

2 A good description of the study setting and hospital changes to accommodate COVID-19 patients has been provided.

3 The sentence on pg. 6 L 133-134 "admission to the ICU after more than 7 days of invasive ventilatory support" needs clarification as invasive ventilation is only possible in ICU.

4 The exclusion of patients who tested PCR positive for COVID-19 but were excluded from the study based on all other exclusion criteria should be clarified. As the flow chart shows, all these patients were recruited from ICU and if the main outcome of the study is survival at 60 days of COVID-19 patients due to all other associated factors, the exclusion of these positive cases would probably cause selection bias in the sample.

5 On pg. 7, L154, an elaboration on how data confidentiality was maintained is required.

6 Detailed information has been provided on data collection methods and variables collected.

Analysis

7 Good explanation on how continuous and binary variables were analyzed.

8 On what basis were variables selected for the multivariable analysis? Please include in the analysis

Results

9 Good presentation of participant selection in flow diagram

10 The tables are well presented with detailed information, however

11 Does the statement on pg10, L209 suggest that no patient dies in ICU? If yes, the outcome variable "death" under ICU outcomes?

12 Vaccination and variants of concern were only possible to be measured for the second wave. I am not sure if they can be included in the analysis and results as this would be a concern of measurement bias.

13 The section on ICU and hospital outcome needs to be more detailed with specific results from table 3 being quoted. Include significant levels too.

14 It is not clear which variables were selected from all the baseline characteristics for assessment of association with survival at 60 days. Table 3 in the supplementary only shows 3 variables, one of which was only measured in the second wave (vaccination) and should not be included as this would bias the results.

Discussion

15 On Pg 17, L 285 it is unclear how the authors conclude that 60-day mortality was 27% less in the second wave. A HR of 0.61 translates to a survival of 39% more in the second wave.

16 Given that this study is comparing waves 1 and 2 and the variant of concerns were only assessed in wave two, the discussion on Gamma variant might not be necessary. However, this can be mentioned as a limitation and a point to note for future comparative studies

17 Overall, the discussion is well tailored balancing the findings with international and national findings from other studies.

6. PLOS authors have the option to publish the peer review history of their article (what does this mean?). If published, this will include your full peer review and any attached files.

Reviewer #1: No

Reviewer #2: No

---

## [Author Response · Author response to Decision Letter 0]

28 Dec 2023

Editorial office

C.1: Please ensure that your manuscript meets PLOS ONE's style requirements, including those for file naming. 

R1: We have adjusted some details of the manuscript and confirmed that it meets the requirements.

C2: Thank you for stating the following in the Competing Interests section… 

R2: We confirm that the statement in the Competing Interests section does not alter our adherence to all PLOS ONE policies on sharing data and materials and we included this recommended statement. Below is the updated Competing Interests statement and we included it in our cover letter.

"Dr. Ferreira reports personal fees from Medtronic, outside the submitted work; Dr. Costa reports personal fees from Timpel, personal fees from Magnamed, outside the submitted work; Dr. Ho reports personal fees from Pan‐American Health Organization, outside the submitted work. This does not alter our adherence to PLOS ONE policies on sharing data and materials. The other authors have no conflict of interest to disclose."

C3: In your Data Availability statement, you have not specified where the minimal data set underlying the results described in your manuscript can be found. PLOS defines a study's minimal data set as the underlying data used to reach the conclusions drawn in the manuscript and any additional data required to replicate the reported study findings in their entirety. All PLOS journals require that the minimal data set be made fully available. For more information about our data policy, please see http://journals.plos.org/plosone/s/data-availability.

Important: If there are ethical or legal restrictions to sharing your data publicly, please explain these restrictions in detail. Please see our guidelines for more information on what we consider unacceptable restrictions to publicly sharing data: http://journals.plos.org/plosone/s/data-availability#loc-unacceptable-data-access-restrictions . Note that it is not acceptable for the authors to be the sole named individuals responsible for ensuring data access.

R3: We would like to update the Data Availability statement in our cover letter. It should now read:

All relevant data are within the paper and its Supporting Information files. Brazilian Data privacy regulations prohibit sharing of individual level data to the public and the ethical approval did not cover public sharing of data for unknown purposes. Anonymized data may be shared upon reasonable request upon contact with the corresponding author (daniela.f@hc.fm.usp.br) or the Research Ethics Committee (cappesq.adm@hc.fm.usp.br).

C4: We note that you have stated that you will provide repository information for your data at acceptance. Should your manuscript be accepted for publication, we will hold it until you provide the relevant accession numbers or DOIs necessary to access your data. If you wish to make changes to your Data Availability statement, please describe these changes in your cover letter and we will update your Data Availability statement to reflect the information you provide.

R4: As mentioned about, we have updated our Data Availability Statement in the cover letter.

C5: Please include captions for your Supporting Information files at the end of your manuscript, and update any in-text citations to match accordingly. Please see our Supporting Information guidelines for more information: http://journals.plos.org/plosone/s/supporting-information. 

R5: Yes, we have adjusted the supporting information files and confirmed that it meets the guidelines. We have added captions for our Supporting Information files at the end of the manuscript, and update in-text citations to match accordingly. 

Responses to the Reviewer #1

Abstract

CR1.1: Consider providing a brief context for readers unfamiliar with the Brazilian healthcare system and the significance of studying outcomes across different waves of the pandemic.

R1: Thank you for your suggestion. We have included a brief context of the Brazilian healthcare system to the abstract (P3 L52) and the methods section (P6 L123).

Introduction

CR1.2: Provide references to the claim made on P5 L101.

R2: We have added references, as suggested (P5 L112). 

Methods

CR1.3: I like how the authors provided a very good description of the study population, highlighting the selection criteria. However, additional information is needed to clarify the specific criteria that were used to define critically ill patients. Detailing parameters like respiratory rate, oxygen saturation, and comorbidities would be helpful for readers to gauge the severity of the cases.

R3: Thank you for your comment. The criteria to define critically ill patients was admission to one of the hospital´s COVID-19 ICUs. ICU admission was regulated by a team of physicians and nurses who received ICU admission requests from the emergency rooms and wards, and outside hospitals. We added this information to the text to improve clarity (P7 L141).

CR1.4: Kindly add references to the Patient Care subsection P7 L143.

On P7 L143, the authors mention the main changes in treatment protocol between 2020 and 2021. Can they also comment on the use of anticoagulants at prophylactic doses to prevent thromboembolism and the nonuse of antibiotics in patients without suspected bacterial infection as reported by Falavigna et al (https://www.jbmede.com.br/index.php/jbme/article/download/69/50).

R4: Thank you for the suggestion, we provided a more detailed description of protocols to the Patient Care subsection and the suggested reference (P7 L158). 

CR1.5: In the Statistical Analysis subsection P8 L180, several essential details are missing that are crucial for a comprehensive understanding of the statistical analysis. The description mentions that Kaplan-Meier curves were plotted for each of the pandemic waves. It's important to clarify if these curves were plotted separately for each group (first wave vs. second wave) or if there was a comparison between the curves. If a comparison was made, the results of the log-rank test, which is commonly used to compare Kaplan-Meier curves between groups, should be reported. 

R5: We agree that the description of the Kaplan-Meier curves and its corresponding statistical analysis was not sufficiently clear. We did perform a logrank test to compare the survival in the two waves in a unadjusted model, and we also used an unadjusted Cox proportional hazards model to estimate the association between pandemic wave and survival, in order to obtain an unadjusted hazard ratio. In our revised manuscript, we report both results (logrank and unadjusted HR). We also modified our statistical analysis section to improve clarity. Changes to the manuscript in P10 L216; P18 L 332; P18 L339; P18 L 347.

CR1.6: Kindly explain the rationale for choosing the variables to be included in the multivariable Cox model-building process. There is no mention of how assumptions such as the proportional hazards assumption in Cox models were checked. It's vital to confirm that these assumptions hold for the validity of the analysis.

R6: We used a conceptual causal diagram in the format of directed acyclic graph (DAG) to estimate the association between the main predictor (admission in the second wave) and the outcome, adjusting for relevant confounders. Variables in the DAG conceptual model were selected based on prior knowledge. We added this information to the methods section (P10 L223), with an appropriate reference (P10 L221) and in the abstract (P3 L54).

We used the Schoenfeld residual method to test the proportional hazards assumption in Cox models. We have added this information to the methods section and bellow we showed the test. Changes to the manuscript in P10 L226.

CR1.7: Information on how the goodness-of-fit of the Cox proportional hazards model was assessed is missing. Techniques like likelihood ratio tests, Akaike Information Criterion (AIC), or Bayesian Information Criterion (BIC) are commonly used to assess model fit. 

R7: Thank you for bringing up this point. Because the specification of our final model was comprised solely of potential confounders previously defined according to our DAG conceptual model, we did not have to select between different models. We also did not include any interaction terms or other higher order terms in our model and did not have to select. We added a sentence to the methods section to clarify this point P10 L227. 

CR1.8: Details about specific packages or functions within R used for the analyses are important for transparency and reproducibility. 

R8: We agree, we used the packages survival and survminer. We added the references to the text (P11 L234). 

Results

CR1.9: I liked the clear presentation of the patients’ characteristics from both waves in Table 1, including demographic information, comorbidities, and other relevant factors summarizing the key differences. This will provide a comprehensive understanding of the patient population studied.

In the Baseline Characteristics subsection P10 L212-219, the authors need to provide specific numerical data from Table 1 to support their statements about the subjects. Instead of using phrases like "male sex was predominant" or "more likely," the authors should directly quote the figures from the table to enhance the precision and clarity of the description. For instance: 

Instead of saying "male sex was predominant in both periods," specify the exact percentages of male patients in both the first and second waves.

Instead of saying "56% were ≥60 years old" and "43% were ≥60 years old," provide the actual numbers of patients in each age group (percentage in brackets) for both waves.

Instead of saying, "Patients in the first wave were more likely to be receiving mechanical ventilation and vasopressors," provide the specific percentages or counts of patients receiving mechanical ventilation and vasopressors in each wave. 

Provide the actual numerical values for mean SAPS3 and SOFA scores in both waves to convey a more precise comparison.

By directly citing the figures from Table 1, readers can have a clear and accurate understanding of the baseline characteristics of the subjects in each wave without having to move back and forth to Table 1.

The same applies to the ventilatory management in the first 24 hours after ICU admission (P12 L225-232) and management in the first 24 hours after ICU admission (P13 L240-247) subsections. See Lalla et al as an example (https://pubmed.ncbi.nlm.nih.gov/35359698/) .

R9: Thank you for the suggestion. We added specific numerical data to the text (P13 L252-258; P14 L277-280; P16 L299-303). We suppressed less relevant details from the text, since they are already shown in the respective tables to avoid excess numerical data repeated in the text. Additionally, we changed the heading of the section on the ventilatory management according to the suggested heading for the section on clinical management (P14 L274). 

CR1.10: On P13 L240, the heading seems a bit vague; the authors can consider changing it to Clinical Management of Patients in the First 24 Hours After ICU Admission.

R10: We agree and changed the heading, and we specified presentation of the Clinical Management of Patients in the First 24 Hours (P15 L294).

CR1.11: In the ICU and hospital outcomes subsection P14 L248-257, similar comments to the "Baseline Characteristics" subsection apply. In addition, include p-values or other indicators of statistical significance to assess whether the observed differences are likely due to chance or represent true disparities between the waves.

R11: Yes, we added numerical data and p-values and once again suppressed less relevant details from the text, to avoid excess repetition (P16 L308-317). 

CR1.12: On P16 L274, the authors mention the multivariable model with vaccination. Adjusting for a variable that is measured in one group but not in another can introduce bias and affect the validity of the results. In this case, adjusting for vaccination as a variable in the Cox model only for the second wave raises some concerns:

(1) Selection Bias: Since vaccination status was only measured in the second wave, it means that the two groups (first wave and second wave) are not directly comparable concerning this variable. The absence of vaccination data in the first wave introduces selection bias, potentially leading to distorted results.

(2) Limitation of generalizability: The results might not be generalizable to populations where vaccination status is differentially available or measured between waves. This limitation could affect the study's broader applicability and relevance.

In the absence of uniform vaccination data, the study could acknowledge this limitation explicitly. Authors should discuss the implications of this limitation on the interpretation of results and potential biases introduced by the unequal measurement of vaccination status between waves.

R12: We share the reviewer’s concerns about the interpretation of the results with vaccination to the DAG model. We debated whether to include it or not to the model. Including vaccination incurs in all the problems mentioned by the reviewer. However, leaving vaccination out of the model might interfere with testing the main hypothesis of the study, given that we were comparing survival between the two waves, and vaccination was available to some patients in the second wave and was expected to reduce mortality. In order to mitigate the risk of bias, we planned and performed a sensitivity analysis without vaccination in the model, and the results of the multivariable model were very similar, but we failed to add it to the original manuscript. It is now added to the methods (P10 L224) and results sections (P18 L339). 

We also agree that the limitations need to be explicitly acknowledged, we added a paragraph about vaccination to the discussion (P21 L401) and modified the limitations paragraph to address this concern (P23 L464 and P24 L473). 

Discussion

CR1.13: The authors fully explored potential reasons behind the observed differences in patient outcomes between the two waves. They discussed factors such as healthcare infrastructure and public awareness. Limitations of the study were mentioned; however, the authors may want to discuss further how these limitations might have influenced the results and interpretations.

R13: We agree and have added a sentence of the impact of the limitations on data interpretation at the end of the paragraph (P24 L473).

Conclusion

CR1.14: Consider discussing the implications of the study findings for clinical practice. How can the insights gained from this study inform future pandemic preparedness and response strategies?

R14: Thank you for your suggestion. We added a sentence about the importance of our findings to show that implementing evidence-based protocols can impact patient outcomes to the last paragraph of the discussion and added additional comments about pandemic preparedness to the conclusion paragraph (P24 L482 and P25 L499). 

Responses to the Reviewer #2

Overall impression and relevance

CR2: The study presents a relevant issue concerning variants of concerns and continuous outbreaks of the COVID-19. It is professionally written, and the methods used are detailed and comprehensive. However, from the topic and aim presented in the introduction, one would expect to see detailed analysis of how the different variants of concern influence survival/mortality in both the first wave and the second wave. But the study presents a comparison of survival/mortality of patients based on admission clinical characteristics during the first 24 hours as its primary outcome and assessing the difference between other clinical outcomes between the 1st and 2nd wave as secondary outcomes. The authors should consider changing the topic and aim to align with the methods and results.

R: Thank you for your comment. Indeed, we were interested in the impact of the emergence of VoCs in the survival of COVID-19 critically ill patients and our hypothesis was that patients admitted in the second wave, which was driven by the emergence of VoCs, had worse clinical outcomes. In the background, we discuss VoCs, but also other factors potentially associated with worse outcomes, such as health care system strain. On the last paragraph of the introduction, we state that our aim was to “compare characteristics, clinical management, and outcomes of critically ill patients hospitalized in the first and second waves of COVID-19 in a large academic hospital in Brazil. We hypothesized that in-hospital mortality would be greater in the second wave.” We did test for VoCs for patients in the second wave. There were missing data, but 94% of the samples detected the Gamma variant. As a result, year of admission, our main predictor of survival in the unadjusted analysis, is a proxy of VoCs since all patients in the first wave had the original strain and almost all patients in the second wave were contaminated with the Gamma variant. Baseline clinical characteristics were added only to the adjusted model to account for confounding, but the main analysis focuses on wave of admission (and therefore, VoCs). Then, we believe that our aims are aligned with the results. In order to better clarify that intention, we modified the title (I believe the reviewer meant title instead of topic), mentioning that we were interested in comparing the two waves, which are driven by different variants (P1 L5).

Background

CR2.1 The statement on pg. 5, L 102-103 requires backing with references.

R1: Thank you for your comment, we added the references (P5 L 112).

Methods

CR2.2 A good description of the study setting and hospital changes to accommodate COVID-19 patients has been provided.

R2: Thank you for your comment.

CR2.3 The sentence on pg. 6 L 133-134 "admission to the ICU after more than 7 days of invasive ventilatory support" needs clarification as invasive ventilation is only possible in ICU. Do you mean admission to the COVID-19 ICU after more than 7 days of invasive ventilatory support? Because invasive ventilation is possible only when a patient is in ICU. Also, why would you exclude these patients unless they were confirmed negative for COVID-19?

R3: We appreciate the comment. However, in our hospital, we sometimes received patients transferred from other hospitals with fewer resources than ours, and occasionally they might have been under mechanical ventilation in the previous hospital for several days and have been received variable ventilatory management, and we believe that they may be very different form our target population, and therefore needed to be excluded. Only 19 patients were excluded for this reason. On the other hand, in our hospital, invasive ventilation was started in the emergency room for patients who arrived with overt respiratory failure, or the wards for patients with sudden deterioration. These patients typically were transferred to the ICU within a few hours and were not excluded from the study. We added a clarification on the methods section (P7 L141 e P7 L148-152). 

CR2.4 The exclusion of patients who tested PCR positive for COVID-19 but were excluded from the study based on all other exclusion criteria should be clarified. As the flow chart shows, all these patients were recruited from ICU and if the main outcome of the study is survival at 60 days of COVID-19 patients due to all other associated factors, the exclusion of these positive cases would probably cause selection bias in the sample.

R4: Thank you for your comment. We aimed to include a very broad and representative sample of critically ill patients with COVID-19 in our study. However, we believe that including patients with certain clinical conditions such as terminal disease and palliative care, for example, or prolonged mechanical ventilation provided in a different hospital, would introduce bias into our sample. Moreover, we also excluded patients who were admitted to the ICU for a reason that was not related to COVID-19 (post exploratory laparotomy for abdominal obstruction, for example), had no COVID-19 symptoms, but tested positive in screening tests, since at the time all patients admitted to our hospital, for any reason, were tested. We edited the methods section to clarify this exclusion criteria (P7 L148-152). Only 4 patients were excluded for this reason and less than 7% of the total sample were excluded for any of the exclusion criteria.

CR2.5 On pg. 7, L154, an elaboration on how data confidentiality was maintained is required.

An elaboration of this should be included especially as online information hacking can occur.

R5: We agree, we added more details (P8 L172).

CR2.6: Detailed information has been provided on data collection methods and variables collected. 

What about this data from patients in the first wave? 

R6: Thank you for your comment. We did not test respiratory samples from patients admitted in the first wave for SARS-CoV-2 variants because at the time, there were no VOCs circulating in Brazil. Patients the first wave we considered infected by the original strain. We added a sentence to the methods section to clarify this point (P9 L200). 

Analysis

CR2.7: Good explanation on how continuous and binary variables were analyzed.

R7: Thank you for your comment.

CR2.8: On what basis were variables selected for the multivariable analysis? Please include in the analysis

R8: Thank you for your comment. As we mentioned in our response to reviewer #1, we used a conceptual causal diagram in the format of directed acyclic graph (DAG) to estimate the association between the main predictor admission in the second wave and the outcome, adjusting for relevant confounders. Variables in the DAG conceptual model were selected based on prior knowledge, as is usually the case. We added this information to the methods section (P10 L 223), with an appropriate reference (P10 L 221).

Results

CR2.9: Good presentation of participant selection in flow diagram

R9: Thank you for your comment.

CR2.10: The tables are well presented with detailed information, however.

R10: We believe this comment precedes the next one.

CR2.11: Does the statement on pg10, L209 suggest that no patient dies in ICU? If yes, the outcome variable "death" under ICU outcomes?

R11: Thank you for pointing out this omission, we added that follow up was continued until discharge, death or transfer (P12 L 246). 

CR2.12: Vaccination and variants of concern were only possible to be measured for the second wave. I am not sure if they can be included in the analysis and results as this would be a concern of measurement bias.

R12: We share both reviewer’s concerns about the interpretation of the results with vaccination to the DAG model. As we mentioned in our response to reviewer #1, we debated whether to include it or not to the model. Including vaccination incurs in all the problems mentioned by the reviewer. However, leaving vaccination out of the model might interfere with testing the main hypothesis of the study, given that we were comparing survival between the two waves, and vaccination was available to some patients in the second wave and was expected to reduce mortality. In order to mitigate the risk of bias, we planned and performed a sensitivity analysis without vaccination in the model, and the results of the multivariable model were very similar, but we failed to add it to the original manuscript. It is now added to the methods (P10 L224) and results sections (P18 L339), and a sentence on the limitations paragraph.

Variants of concern were not used as predictors in the survival analysis, we used it to characterize our study population on the second wave. 

CR2.13: The section on ICU and hospital outcome needs to be more detailed with specific results from table 3 being quoted. Include significant levels too.

R13: We agree, we added numerical data and p-values.

CR2.14: It is not clear which variables were selected from all the baseline characteristics for assessment of association with survival at 60 days. Table 3 in the supplementary only shows 3 variables, one of which was only measured in the second wave (vaccination) and should not be included as this would bias the results.

R14: We agree that the original manuscript did not explicitly mentioned the criterion for selecting variables for the survival model. We used a conceptual causal diagram in the format of directed acyclic graph (DAG) to estimate the association between the main predictor (admission in the second wave) and the outcome, adjusting for relevant confounders. Variables in the DAG conceptual model were selected based on prior knowledge, as is usually the case. We added this information to the methods section (P10 L 223), with an appropriate reference (P10 L 221) and in the abstract (P3 L54). The discussion about the inclusion of vaccination in the model and the sensitivity analysis is discussed above. 

Discussion

CR2.15: On Pg 17, L 285 it is unclear how the authors conclude that 60-day mortality was 27% less in the second wave. A HR of 0.61 translates to a survival of 39% more in the second wave.

R15: Thank you for your comment. We agree that the sentence is confusing because it mentions mortality instead of survival. All of our analyses are based on survival, not mortality, so mentioning the mortality rate on the first paragraph of Discussion was inappropriate. Changes to the manuscript in P19 L354.

CR2.16: Given that this study is comparing waves 1 and 2 and the variant of concerns were only assessed in wave two, the discussion on Gamma variant might not be necessary. However, this can be mentioned as a limitation and a point to note for future comparative studies.

R16: Thank you for your comment. We agree and added a sentence about future comparative studies to the Discussion section (P22 L421). 

CR2.17: Overall, the discussion is well tailored balancing the findings with international and national findings from other studies.

R17: Thank you for your comment.

---

## [Decision Letter · Decision Letter 1]

15 Feb 2024

Temporal trends of severity and outcomes of critically ill patients with COVID-19 after the emergence of variants of concern: A comparison of two waves

PONE-D-23-21314R1

Dear Dr. Freitas,

We’re pleased to inform you that your manuscript has been judged scientifically suitable for publication and will be formally accepted for publication once it meets all outstanding technical requirements.

Kind regards,

Luis Felipe Reyes, M.D., Ph.D., MSc.

Academic Editor

PLOS ONE

Additional Editor Comments (optional):

Reviewers' comments:

Reviewer's Responses to Questions

**Comments to the Author**

1. If the authors have adequately addressed your comments raised in a previous round of review and you feel that this manuscript is now acceptable for publication, you may indicate that here to bypass the “Comments to the Author” section, enter your conflict of interest statement in the “Confidential to Editor” section, and submit your "Accept" recommendation.

Reviewer #1: All comments have been addressed

Reviewer #2: All comments have been addressed

2. Is the manuscript technically sound, and do the data support the conclusions?

Reviewer #1: Yes

Reviewer #2: Yes

3. Has the statistical analysis been performed appropriately and rigorously? 

Reviewer #1: Yes

Reviewer #2: Yes

4. Have the authors made all data underlying the findings in their manuscript fully available?

Reviewer #1: No

Reviewer #2: No

5. Is the manuscript presented in an intelligible fashion and written in standard English?

Reviewer #1: Yes

Reviewer #2: Yes

6. Review Comments to the Author

Reviewer #1: Thank you for allowing me the opportunity to review the revised manuscript on "Temporal trends of severity and outcomes of critically ill patients with COVID-19 after the emergence of variants of concern: A comparison of two waves." I appreciate the authors' efforts in addressing the major comments and concerns raised in the initial submission.

The methods section now provides a comprehensive description of the statistical methods employed, addressing previous deficiencies.

The results section has been reframed and rewritten in accordance with expected reporting guidelines. This is commendable.

Regarding the discussion section, I noticed a shift from discussing "survival" on line 322 to using the term "mortality" on line 343. To maintain consistency and avoid confusion for readers, I recommend continuing with the term "survival" throughout this section.

Overall, the manuscript is now well-structured and reads smoothly.

Reviewer #2: All coments have been adressed appropraitly by the authors.

The authors have re-analysed their data and provided more detailed explanantions and transparency.

7. PLOS authors have the option to publish the peer review history of their article (what does this mean?). If published, this will include your full peer review and any attached files.

Reviewer #1: No

Reviewer #2: No

---

## [Editor Report · Acceptance letter]

27 Feb 2024

PONE-D-23-21314R1 

PLOS ONE

Dear Dr. Freitas, 

I'm pleased to inform you that your manuscript has been deemed suitable for publication in PLOS ONE. Congratulations! Your manuscript is now being handed over to our production team.

Kind regards, 

on behalf of

Dr. Luis Felipe Reyes 

Academic Editor

PLOS ONE